# Solvers for the Hermitian and the pseudo-Hermitian Bethe-Salpeter equation in the Yambo code: implementation and performances

Petru Milev[1][⋆]⊙, Blanca Mellado-Pinto[2]⊙, Muralidhar Nalabothula[3]⊙, Ali Esquembre-Kučukalić[4]⊙, Fernando Alvarruiz[2]⊙, Enrique Ramos[2]⊙, Alejandro Molina-Sanchez[4]⊙, Ludger Wirtz[3]⊙, Jose E. Roman[2]⊙, and Davide Sangalli[1][†]⊙

**1** Istituto di Struttura della Materia-CNR (ISM-CNR), Area della Ricerca di Roma 1, Monterotondo Scalo, Italy

**2** D. Sistemes Informàtics i Computació, Universitat Politècnica de València, Camí de Vera s/n, 46022, València, Spain

**3** Department of Physics and Materials Science, University of Luxembourg, 162a avenue de la Faïencerie, L-1511 Luxembourg, Luxembourg

**4** Institute of Materials Science (ICMUV), University of Valencia, Catedrático Beltrán 2, 46980, Valencia, Spain

⋆ petru.milev@ism.cnr.it , † davide.sangalli@ism.cnr.it

## Abstract

We analyze the performance of two strategies in solving the structured eigenvalue problem deriving from the Bethe-Salpeter equation (BSE) in condensed matter physics. The BSE matrix is constructed with the `Yambo` code, and the two strategies are implemented by interfacing `Yambo` with the ScaLAPACK and ELPA libraries for direct diagonalization, and with the SLEPc library for the iterative approach. We consider both the Hermitian (Tamm-Dancoff approximation) and pseudo-Hermitian forms, addressing dense matrices of three different sizes. A description of the implementation is also provided, with details for the pseudo-Hermitian case. Timing and memory utilization are analyzed on both CPU and GPU clusters. Our results demonstrate that it is now feasible to handle dense BSE matrices of the order of $10^5$.

## Contents

---

# 1   Introduction

The Bethe-Salpeter Equation (BSE) is the state-of-the-art first-principles approach to compute neutral excitations in material science [1, 2]. It explicitly accounts for electron–hole interactions, allowing for an accurate description of excitonic effects and thereby enabling reliable prediction of optical properties in a wide range of materials, including 2D materials [3, 4], wide band-gap insulators and transition metal oxides [5, 6], semi-conductors, and magnetic materials [7, 8]. It was also recently used to compute magnons in magnetic materials [9]. It can be formulated as a non-Hermitian structured eigenvalue problem given by

$$\begin{pmatrix} R & C \\ -C^\dagger & -A \end{pmatrix}\begin{pmatrix} X_I \\ Y_I \end{pmatrix} = \omega_I \begin{pmatrix} X_I \\ Y_I \end{pmatrix} \tag{1}$$

where $R$ and $A$ are dense Hermitian matrices, $R = R^\dagger$ and $A = A^\dagger$, while $C$ is a dense rectangular matrix. Here $R^\dagger = (R^*)^T$ denotes conjugate transposition. We further define the $M$ and $\Omega$ matrices as

$$\Omega = \begin{pmatrix} I & 0 \\ 0 & -I \end{pmatrix}, \quad M = \begin{pmatrix} R & C \\ C^\dagger & A \end{pmatrix}, \tag{2}$$

which allows us to rewrite Eq. (1) as

$$\Omega M \begin{pmatrix} X_I \\ Y_I \end{pmatrix} = \omega_I \begin{pmatrix} X_I \\ Y_I \end{pmatrix}. \tag{3}$$

We refer to the full BSE matrix as $H^{2p}$, and to its structure as pseudo-Hermitian [10–13]. A matrix is said to be pseudo-Hermitian, if there exists a Hermitian linear operator $\eta$, such that $H^\dagger = \eta H \eta^{-1}$ [14]. In the case of BSE, the matrix $H^{2p}$ satisfies the condition with respect to the $\eta = \Omega$, and is thus $\Omega$-pseudo-Hermitian. The BSE Hamiltonian, as shown in Eq. (3), can be expressed as the matrix product of two Hermitian matrices. If the $M$ matrix is positive definite, which can be verified by Cholesky factorization, then the BSE problem reduces to a Hermitian-definite generalized eigenvalue problem with real eigenvalues [10]. Otherwise, it should be treated as a general non-Hermitian eigenvalue problem. Additionally, it can be shown that BSE matrix has a block containing transitions of positive energy (resonant matrix $R$), a block containing transitions of negative energy (anti-resonant matrix $A$), and two blocks which mix positive and negative energy transitions (coupling matrices $C$ and $C^\dagger$). Neglecting the coupling term, i.e., setting $C = 0$, corresponds to the Tamm-Dancoff approximation (TDA) [1, 15]. Regardless of $M$ being positive definite or not, the BSE matrix in Eq. (1) must have a real eigenspectrum to describe a physical system, provided that lifetimes effects are not included, i.e., all the matrices are frequency independent, as it is usually the case. The structure of the

BSE also allows further simplifications of the eigenvalue problem in the special case where R = A, as discussed later. This enables more efficient solution methods that go beyond the standard Hermitian-definite generalized eigenvalue approach [16].

The same structure also appears in the eigenvalue problems derived within Time-Dependent Density Functional Theory (TD-DFT) and within the Random-Phase Approximation (RPA).

The BSE eigenvalue problem is coded in many condensed-matter (and quantum chemistry) packages, both open-source, such as Yambo [17, 18], BerkeleyGW [2, 19], Abinit [20, 21], Exc [22], Exciting [13, 23, 24], GPAW [25, 26], Simple [27], MolGW [28] and commercial or private, such as VASP [12, 29], TurboMole [30], and Fiesta [31, 32]. Considering also TD-DFT and RPA, the number of scientific codes implementing the eigenvalue problem of Eq. (1) becomes very large, including applications in quantum chemistry and nuclear physics. Most of them use the Hermitian-definite generalized eigenvalue solvers implemented in eigensolver packages. In condensed matter, most BSE applications are done within the TDA, where the BSE eigenvalue problem becomes Hermitian. Indeed, at variance with the case of isolated systems studied in quantum chemistry, the BSE can be recast into a Hermitian quadratic problem only in presence of time-reversal symmetry [12]. However, there are physical cases where going beyond the TDA is needed, and time-reversal is not available [33]. Thus, the full pseudo-Hermitian problem must be addressed [10, 11, 13].

For the case of optical properties, we have $A = R^*$, while $C = C^T$. The size of both $R$ and $C$ is $N \times N$. Defining $N_{\mathbf{k}}$ as the number of k-points in the full Brillouin zone, $N_v$ as the number of included occupied (valence) states, and $N_c$ as the number of unoccupied (conduction) states, $N = N_{\mathbf{k}} N_v N_c$ is the total number of optical transitions. Numerically, the most demanding step is the building of the $R$ and $C$ matrices. This operation scales as $\mathcal{O}(N^2)$, with a large prefactor that depends on the square of the basis-set size. For plane-wave codes, this corresponds to $\mathcal{O}(N_G^2)$ where $N_G$ is the number of G-vectors. This is why, with the advent of HPC machines and complex structures, a significant effort has been devoted to improve the matrix generation algorithm, taking advantage of MPI, OpenMP and GPU porting schemes such as CUDA Fortran, OpenMP5 or OpenACC. Thanks to these developments, matrices with a size of the order $N \propto 10^5$ can be easily generated. Even larger matrices can be generated by means of interpolation techniques [34, 35].

The solver step has received significantly less attention, since the computational time needed is usually lower. However, if the solver implementation is not optimized, it can become a barrier, preventing the solution of the eigenvalue problem. The issue is particularly severe when the excitonic wave-functions, i.e., the eigenvectors of the BSE, need to be analyzed. The full diagonalization has a scaling $\mathcal{O}(N^3)$, and would become the most demanding part of the calculation at a certain size. Iterative schemes show a better scaling, typically $\mathcal{O}(N^2)$. In both approaches, parallel and GPU ported solvers are needed to handle large matrices. Another issue is the storing of the BSE matrix, either on disk (taking advantage of parallel I/O) or directly into RAM memory by distributing it over several nodes or GPU cards. In other eigenvalue problems in condensed matter, the matrix can be easily stored in memory and/or even generated on the fly when using iterative solvers, since (i) the generation process is much faster, and (ii) many matrix elements can be related to each other. The main example in condensed-matter physics is the eigenvalue problem of DFT, where, as opposed to BSE, most efforts have been devoted to develop efficient and highly parallelized iterative algorithms for the solver step, allowing to reach sizes of the order of $10^7$ which are not feasible in BSE. In the BSE instead, explicit storage is needed even if iterative solvers are adopted. Also accounting for some memory redundancy used by the available algorithms, this can easily lead to exceeding the barrier of 1 TB of data to be handled. In the most demanding cases, even tens of TB for a single simulation are required.

The size of the Bethe–Salpeter Equation (BSE) matrix plays an important role in the study

of excitons and the simulation of optical properties. In general, using the full BSE matrix with as many bands as possible is desirable in order to properly converge excitonic energies and to capture exciton physics within a material. More specifically, matrices with size of the order $10^4$ - $10^5$ can be easily reached when studying 2D heterostructures [36, 37], systems with defects [38, 39], system that involves molecular interactions such as functionalized materials (e.g., functionalized graphene), or systems containing impurities. In some cases, neglecting the coupling term is not an option. For instance, the coupling term is known to be essential when studying localized states which are important in some of the applications mentioned above. Moreover, when moving to large-scale systems, such as supercells or nanostructures, the BSE matrix becomes extremely large. In such cases, efficient diagonalization algorithms are necessary to make the calculations computationally feasible.

In the present work, we analyze the performances of BSE solvers, two based on direct diagonalization, and the other based on an iterative scheme, and we use them to solve prototype BSE matrices. The two approaches have been implemented in the `Yambo` code, a plane-wave code, taking advantage of the interfaces with libraries such as ScaLAPACK [40], ELPA [41–43], PETSc [44] and SLEPc [45]. We consider both the Hermitian BSE within the TDA, and the full structured pseudo-Hermitian BSE matrix. For the latter case, we discuss in detail how we take advantage of the pseudo-Hermitian structure in both solvers to reduce the computational time. We provide parallel performances, using a local cluster for CPU only simulations, and the Leonardo machine for GPU ported simulations. An analysis of the memory used on the host is also provided.

## 2 Description of the Algorithm

In this section we provide details about the two computational approaches: full diagonalization and iterative solver.

### 2.1 The Exact Diagonalization

#### 2.1.1 The Diagonalization Scheme for Full BSE Matrix (Coupling Case)

We briefly outline the special algorithm for diagonalizing the BSE Hamiltonian beyond TDA given in Eq. (1) when $A = R^*$ and $C = C^T$, as shown in Ref. [16]. The first step in solving the BSE Hamiltonian, $H^{2p}$, with this special structure is to construct a real symmetric matrix $M_r$ of the same dimension, which satisfies the relation $Q^\dagger H^{2p} Q = -iJM_r$, where $J$ is a real skew-symmetric matrix and $Q$ is a unitary matrix given by:

$$Q = \frac{1}{\sqrt{2}} \begin{pmatrix} I & -iI \\ I & iI \end{pmatrix}, \quad J = \begin{pmatrix} 0 & I \\ -I & 0 \end{pmatrix}. \tag{4}$$

The expression for $M_r$ is given by:

$$M_r = \begin{pmatrix} \mathrm{Re}(R+C) & \mathrm{Im}(R-C) \\ -\mathrm{Im}(R+C) & \mathrm{Re}(R-C) \end{pmatrix}. \tag{5}$$

If $M$ is positive definite, then $M_r$ is also positive definite [16], and we can perform a Cholesky decomposition on the real symmetric matrix $M_r$, i.e., $M_r = LL^T$ where $L$ is a real lower triangular matrix. This allows us to cast the original eigenvalue problem into the following form:

$$-iWL^T \begin{pmatrix} \bar{X}_I \\ \bar{Y}_I \end{pmatrix} = \omega_I L^T \begin{pmatrix} \bar{X}_I \\ \bar{Y}_I \end{pmatrix} \tag{6}$$

where $W = L^T J L$ is a real skew-symmetric matrix, and $-iW$ is Hermitian, having the same eigenvalues as the original BSE Hamiltonian. In Eq. (6), the eigenvalues of $W$ differ from the $H^{2p}$ eigenvalues by a factor of $-i$. The eigenvalues of a skew-symmetric matrix come in pairs and are purely imaginary, the eigenvalues of $H^{2p}$ are real and occur in pairs, i.e., $(-\omega_I, \omega_I)$. Therefore, the entire BSE problem can be viewed as a real skew-symmetric eigenvalue problem.

In diagonalization, the tridiagonalization step is the computationally expensive part. The algorithm described here enables the tridiagonalization of a real matrix instead of a Hermitian one, significantly reducing both floating-point operations and storage requirements. Once the tridiagonal form of the skew-symmetric matrix is obtained, multiplying the result by $-i$ allows the use of Hermitian tridiagonal solvers to compute the eigenvectors of $W$ [43,46]. After computing the eigenvectors, a back-transformation is applied to retrieve those of the BSE matrix. Moreover, the left eigenvectors for positive eigenvalues, and the left/right eigenvectors corresponding to its negative partner can be determined without explicit computation [16]. More details on the implementation can be found in Ref. [47]. Consequently, we only need to obtain the right eigenvectors associated with positive eigenvalues. The left and right eigenvectors of the positive and negative eigenvalues for the BSE matrix are related as follows [16]:

$$
\begin{pmatrix}
\text{Right eigenvector for } \omega_I & \text{Left eigenvector for } \omega_I \\
\begin{pmatrix} X_I \\ Y_I \end{pmatrix} & \begin{pmatrix} X_I \\ -Y_I \end{pmatrix} \\
\text{Right eigenvector for } -\omega_I & \text{Left eigenvector for } -\omega_I \\
\begin{pmatrix} Y_I^* \\ X_I^* \end{pmatrix} & \begin{pmatrix} -Y_I^* \\ X_I^* \end{pmatrix}
\end{pmatrix}
\tag{7}
$$

where $\omega_I > 0$.

Unlike Hermitian-definite generalized eigenvalue solvers, this method reduces floating-point operations for the most computationally demanding steps. Furthermore, when a direct Hermitian-definite generalized eigenvalue solver is used on the BSE matrix, it can destroy the special properties of this matrix and potentially break degeneracies [16], which is highly undesirable in these types of calculations. Aside from improving efficiency, this approach is also robust to such errors.

### 2.1.2 Interface with ScaLAPACK and ELPA

To efficiently diagonalize the excitonic Hamiltonian in `Yambo`, both in the Hermitian case and, for the coupling case, using the algorithm described above, an MPI redistribution from the `Yambo` parallel structure to a block-cyclic layout is performed in two main steps. First, the matrix elements are locally grouped along with their corresponding global indices based on their destination rank. Then, a single `MPI_AlltoAllv` operation is used to transfer the data to their respective destination according to the block-cyclic scheme. Once the matrix is distributed correctly, standard functions from linear algebra libraries can be applied for efficient computations. Here we use ScaLAPACK and ELPA. For the TDA case, i.e., when the BSE matrix is Hermitian, we use standard Hermitian eigensolvers. ELPA provides two types of Hermitian solver, with type-2 set as the default in `Yambo`. In ScaLAPACK, the `p?heevr` solver is used. Both solvers allow one to compute the lowest part of the spectrum. One advantage of ELPA is that its newer versions are specifically optimized for skew-symmetric matrices [43]. This makes ELPA particularly well-suited for solving pseudo-Hermitian or skew-symmetric eigenvalue problems. Implementing such solvers with ELPA is more straightforward, whereas using ScaLAPACK for the same purpose would require additional modifications to handle the skew-symmetric structure. However, ScaLAPACK enables the extraction of eigenvectors within a specified range of eigenvalues or indices.

175 Since ScaLAPACK does not provide a dedicated solver for real skew-symmetric matrices,
176 the tridiagonalization routines designed for symmetric matrices can be slightly modified to
177 accommodate them, as both share similar computational structures. For the present imple-
178 mentation, we employ ScaLAPACK's Hermitian solver by passing the matrix $-iW$, which con-
179 verts the real skew-symmetric matrix $W$ into a Hermitian form. In future releases, we plan
180 to modify ScaLAPACK's real tridiagonalization routines to directly handle the skew-symmetric
181 matrix $W$, thereby eliminating the need for this transformation.

182 The interface described above has recently been implemented through the development
183 of a new library [47], which is interfaced via the Yambo code. With this new implementation
184 of the diagonalization solver, capable of computing either the full or partial eigenspectrum,
185 we address the long-standing bottleneck related to the diagonalization of BSE matrices in the
186 Yambo code. Although the code is not yet available in the official Yambo release, it is freely
187 distributed through a recently created fork of the code called Lumen [48, 49].

## 2.2 The Iterative Solver

189 Yambo offers two possible approaches: the first is to use a Lanczos iteration with quadra-
190 ture rules to obtain the optical absorption spectrum directly, without explicitly computing the
191 eigenpairs. The second approach is to use SLEPc to obtain the eigenpairs, as the eigenvectors
192 provide additional information to compute the wave functions. In many cases, a few eigen-
193 values are sufficient to calculate the optical absorption spectrum with enough accuracy, so an
194 iterative method can be used.

### 2.2.1 The SLEPc Solver for the Full BSE Matrix (Coupling Case)

196 Prior to the release of SLEPc 3.22, the eigenproblem was solved with a standard non-Hermitian
197 two-sided eigensolver, with the possibility to use shift-and-invert to accelerate the convergence
198 of the desired eigenvalues, which are those closest to zero, and lie in the innermost part of
199 the spectrum because of its symmetry. Since the release of SLEPc 3.22, three new methods
200 are available based on [50, 51], which are three variations of a structure-preserving Lanczos
201 iteration with a thick restart.

202 Essentially, the problem is projected into a smaller structured problem by building a spe-
203 cific Krylov vector basis, which also has a block structure because of the choice of the initial
204 vector. The basis vectors need to be computed accurately and stored in order to obtain the
205 approximate eigenvectors. To prevent the basis from growing too large, a thick restart is used,
206 taking care not to destroy the structure. These methods exploit the structure and properties
207 of the matrix both numerically (symmetry of the matrix blocks and the spectrum) and com-
208 putationally, working implicitly with the lower blocks of the matrix and vector basis, without
209 storing them explicitly. All three methods are very similar in terms of operations. We focus
210 our performance analysis on the Shao method, which is the current default solver in SLEPc for
211 BSE problems.

212 The Shao method does not require the use of the computationally expensive shift-and-
213 invert technique because it computes the eigenvalues of $H^{2p}$, folding the spectrum so that the
214 internal eigenvalues become external and converge first. A last step is required to obtain the
215 positive eigenvalues and associated eigenvectors of $H^{2p}$. The negative eigenvalues and their
216 eigenvectors, and the left eigenvectors, can be obtained trivially from them. The details of the
217 SLEPc solvers for the Bethe–Salpeter eigenproblem are available in [52].

### 2.2.2 MPI Distribution and GPU Porting in SLEPc

In the interface between `Yambo` and SLEPc two different schemes are available. The first, or *shell* approach, is based on *shell matrices*. These are matrices that do not store their values explicitly but instead define how operations on them are carried out. This allows the solver to reuse the existing `Yambo` data and parallel structures directly. Because `Yambo` efficiently stores the BSE matrix in memory by exploiting its symmetry properties, this approach requires less memory. However, it comes at the cost of relying on a parallel scheme that is optimized for building of the BSE matrix rather than for solving the eigenvalue problem. Additionally, this approach does not support GPU acceleration. The second approach, called the *explicit* approach is based on a redistribution of the BSE matrix from the block parallel `Yambo` structure to the SLEPc structure. In SLEPc, MPI parallelization is based on standard one-dimensional row-wise partitioning of matrices and vectors. This is the scheme used by PETSc, which is the library on which SLEPc is built. In the case of GPU simulations, the underlying local computations are offloaded to the NVIDIA GPU using CUDA (or HIP for AMD GPUs). The GPU porting of the `Yambo`-SLEPc interface was achieved by defining the redistributed matrix directly on GPU, specifically using the SLEPc matrix type `MATDENSECUDA`. As in the diagonalization case, this new algorithm is not yet included in the official `Yambo` release, but it is freely available through a recently developed fork of the code called Lumen [48].

The most expensive step of the algorithm are the matrix-vector products that generate the new vectors of the basis. These vectors need to be generated one by one, as they have to be orthogonalized against the previous ones. Additionally, although the solver is also optimized in the case of a sparse matrix, the problems that arise from applications are typically dense, resulting in an expensive matrix-vector product.

## 3 Results and discussion

### 3.1 Details on the Performed Simulations

The characteristics of two clusters used for CPU-only and GPU simulations are reported in Table 1. We constructed matrices from Eq. (1) in single precision (using 32-bit float numbers for both real and imaginary parts) for the case $A = R$ for a single layer of chromium triiodide $CrI_3$. The set of BSE-like matrices considered in the present report are generated by replacing the demanding BSE kernel with the cheaper TD-DFT kernel [1, 53], as implemented in the `Yambo` code. Specifically, the matrix blocks are defined as follows:

$$R_{i,i'} = \Delta\epsilon_i \delta_{i,i'} + v_{i,i'} + f_{i,i'}^{xc}, \tag{8}$$

$$C_{i,j'} = v_{i,j'} + f_{i,j'}^{xc}. \tag{9}$$

Here, $i = \{nm\mathbf{k}\}$ is a transition index. In the resonant block both $i$ and $i'$ refer to positive energy transitions from occupied to empty states, $n\mathbf{k} \to m\mathbf{k}$, while in the coupling block $j'$ refers to negative energy transitions, $m\mathbf{k} \to n\mathbf{k}$, and $\Delta\epsilon_i = \epsilon_{m\mathbf{k}} - \epsilon_{n\mathbf{k}}$ are energy differences. We used an energy cut-off of 4 Ry for the eh-exchange $v$, and 10 Ry for the $f_{xc}$ kernel. These parameters determine the prefactor for the building time of the BSE kernel which we do not analyze here. The size of the BSE matrix is defined by the number of bands used. We considered the three cases reported in Table 2. Here we focus on the performance of the two solvers described in Sect. 2. For direct diagonalization, ScaLAPACK and ELPA libraries (solver type 2 on CPU, type 1 on GPU) were used with a block size of 64 × 64 for the block-cyclic distribution. For the iterative solver, the SLEPc library was used, with either *explicit* (e) or *shell* (s) matrix formats. In our experience, when the eigenvectors are extracted, if more than 5% of the spectrum

Table 1: Characteristics of the clusters used to perform the simulations.

|  | ISM-CNR Cluster | Leonardo Booster |
|---|---|---|
| CPU model | Intel Xeon Gold 5218 | Intel Xeon Platinum 8358 |
| CPU Max Freq. | 2.30GHz | 2.60GHz |
| Nodes used | Up to 3 | Up to 32 |
| Cores per node | 32 | 32 |
| Memory per node | 86406 [MB] | 494000 [MB] |
| Node interconnect | ConnectX®-4 VPI Adapter Card, FDR IB (56Gb/s) | 2x dual-port Infiniband HDR interconnect network interface (400 Gbps aggregated) |
| Operating system | CentOS Linux Release 7.7.1908 | Red Hat Enterprise Linux Release 8.7 (Ootpa) |
| GPUs per Node | – | 4 |
| model | – | NVIDIA A100 GPU HBM2e |
| architecture (c.c.) | – | Ampere (8.0) |
| memory | – | 64 [GB] |
| perf. (S.P.) | – | 75000 [Gflops/node] |
| CUDA version | – | 11.8 |
| Fortran compiler | GFortran 10.2.1 | NVFortran 24.9 |
| MPI | OpenMPI-4.1.1 | OpenMPI-4.1.5 |
| BLAS/LAPACK | 3.12 | OpenBLAS-0.3.24 |
| ScaLAPACK | 2.2.1 | 2.2.0 |
| ELPA | 024.05.001 | 024.03.001 |
| SLEPc | 3.22.0 | 3.22.1 |
| Yambo | 5.3.0 | 5.3.0 |

is required, direct diagonalization routines are recommended. For smaller fractions of the spectrum, the SLEPc iterative solvers are preferable. Here we fix the number of extracted eigenvectors to 100. In the parallel runs this ranges from 1% (for the smallest matrices), to 0.1% (for the largest considered matrix) of the spectrum. We selected a fixed number of 100 eigenvalues because, when using the SLEPc solver, the primary interest typically lies in the characterization of the lowest excitonic peak. In practice, this number is more than sufficient to capture the relevant low-energy excitations. If additional eigenvectors are required, for instance, to investigate higher excitonic peaks or a broader portion of the spectrum, then one must resort to full diagonalization solvers, which provide access to the complete set of excitonic states.

In order to analyze the performances of the simulations, we define the following quantities. $N_i$: number of rows in the resonant block of the matrix for the run $i$. $n_i$: number of MPI tasks for the run $i$. $t_i$: total simulation time for the solver step for the run $i$. $e_i(j) = \frac{n_j t_j}{n_i t_i}$: efficiency of the run $i$ with $n_i$ MPI tasks w.r.t. the run $j$ with $n_j$ MPI tasks. $M_i$: the peak memory or

Table 2: Relationship between memory, matrix size and BSE bands for resonant (res.) and coupling (cpl.) matrices. Memory estimated assuming 64-bit single precision floating point complex numbers.

| BSE bands used | R block size | Est. Mem. [GB] | | Type of Calculation |
|---|---|---|---|---|
|  |  | res. case | cpl. case |  |
| 31-40 | 10368 | 0.80 | 1.60 | CPU & GPU |
| 23-40 | 36288 | 9.81 | 19.62 | CPU & GPU |
| 23-48 | 103680 | 80.09 | 160.18 | GPU only |

maximum resident memory on the host RAM, obtained by summing over the peak memory of all MPI tasks, for the run $i$. We will show, both for CPU and GPUs runs:

- Time complexity plots, i.e., total simulation time ($t_i$) on a single CPU vs matrix size ($N_i$).

- Strong scaling plots, i.e., time ($t_i$) vs number of MPI tasks ($n_i$) at fixed matrix size $N_i$.

- Memory scaling plots, i.e., host memory ($M_i$) vs number of MPI tasks ($n_i$) at fixed matrix size ($N_i$).

- Efficiency plots, i.e., efficiency ($e_i(j)$) vs number of MPI tasks ($n_i$) at fixed matrix size. For the CPU case $n_j = n_{\text{CPU}} = 4$, while for the GPU case $n_j = n_{\text{GPU}} = 2$.

For an efficiency plot, when a reference value is not available, we estimate it using the exponential fit, $t_i = t_1 n_i^p$, where $t_1$ represents the time required for a run on a single MPI task, and $p$ is a constant that characterizes how the simulation time scales with the number of MPI tasks [54].

For perfect scaling, $p = -1$, i.e., doubling the number of MPI tasks should halve the simulation time. Any value $p > -1$ indicates inefficiencies in parallelization, with a time reduction lower than expected. We will refer to the fitted parameter $p$ as the *parallelization factor*.

For both CPU and GPU computations, we measured memory usage using the Linux `time` command, which reports peak memory on host for each MPI task. For some of the runs we also monitored the peak memory of the run reported by the PETSc internal memory tracking tools. To analyze the memory use, we assume that the allocated host memory is due to arrays/matrices which are either perfectly distributed or fully duplicated. We define the distributed memory as the intercept, and the duplicated memory as the slope of the linear fit of the allocated memory vs number of MPI tasks. Moreover, since in some simulations we observe a sudden memory jump from the serial to the parallel simulations, we also define the parallel–overhead memory as the difference between the fit intercept and the memory allocated in the serial run. We attribute this additional memory increase in parallel runs to the allocation of extra variables not present during serial execution. Memory extracted by the `time` command and the PETSc tracker agree well, with a parallel overhead memory detected only by the `time` command. [55]

The conclusions presented in this work are expected to be largely independent of the specific material. $CrI_3$ is employed solely as a representative case for generating a dense BSE matrix. The only aspect that can vary from case to case is the convergence criteria. The diagonalization algorithm scales as $N^3$ and the time to solution does not depend on the matrix properties. For the iterative SLEPc solver, the time required for a single iteration is also independent of the matrix properties, while the number of iterations needed might be affected by the matrix properties. For instance, if the matrix has clustered eigenvalues, this will hinder convergence, resulting in a larger number of iterations. However, this does not usually occur in BSE applications, and the chosen matrix gives a good representation of the needed time to solution for a wide range of cases. Apart from this, once the matrix is constructed, the diagonalization procedure itself is general and does not depend on the underlying material.

## 3.2 Comparison of non-Hermitian vs pseudo-Hermitian Algorithms

Before analyzing the CPU and GPU scaling of the simulations, we first analyze how the computational time scales with matrix size (i.e., time complexity) using a single CPU. We compare the standard algorithms for Hermitian (H) matrices (corresponding to resonant case), standard algorithms for non-Hermitian matrices (NH), and novel algorithms that exploit the pseudo-Hermitian (PH) structure of the coupling matrix. Here we consider three different solvers: (i) full diagonalization which is based on (Sca)LAPACK [56] (see Sect. 2.1.1), (ii) Krylov-Schur

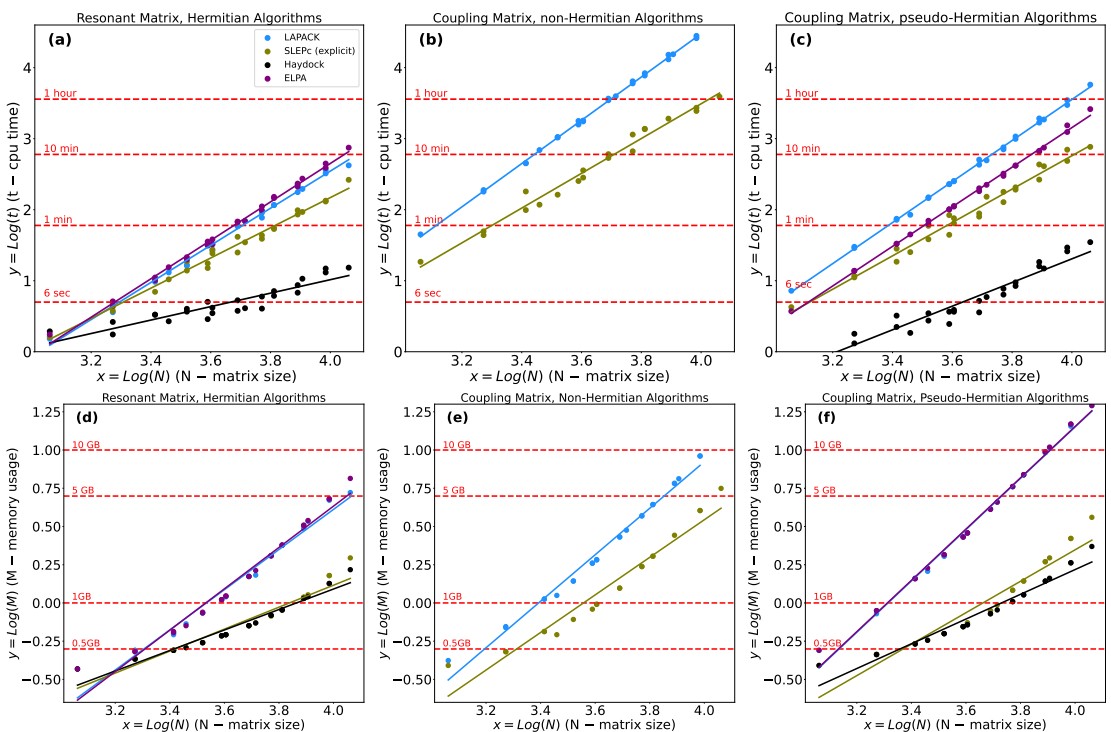

Figure 1: Time complexity (panels (a)–(c)) and memory complexity (panels (d)–(f)) for solving the BSE eigenvalue problem using a single CPU. Results are shown for: (a,d) the Hermitian case, (b,e) the coupling case with a non-Hermitian algorithm, and (c,f) the coupling case with a pseudo-Hermitian algorithm. In all cases, $N$ denotes the size of the resonant block. Simulations were carried out on the ISM cluster.

iterative solver based on the SLEPc library (see Sect. 2.2), and (iii) Haydock iterative solver. The Haydock solver performance is reported for reference as an example of an iterative algorithm implemented in `Yambo`, which already exploits the pseudo-Hermitian structure of the matrix [51].

Fig. 1 presents time complexity plots, and Table 3 reports the corresponding slope and intercept values, $log(t) = A \cdot log(N) + B$. By moving from a logarithmic representation to the direct time versus matrix size relation, the *slope* A corresponds to the scaling exponent of the algorithm with respect to matrix size. Lower values of A indicate better scaling. The *intercept* B represents the computational prefactor, which is also more favorable when smaller. The relation is expressed as $t = B N^A$. The expected scaling is $\mathcal{O}(N^2)$ for SLEPc and $\mathcal{O}(N^3)$ for exact diagonalization. Fig. 1 shows that full diagonalization is the slowest algorithm, while Haydock is the fastest, and SLEPc is intermediate. The performance differences between SLEPc and Haydock arise from the fact that the Haydock algorithm only extracts the optical spectrum, while the SLEPc algorithm also extracts a few eigenvalues and eigenvectors [18].

By comparing panels (b) and (c) of Fig. 1, we observe a significant reduction in the time to solution when switching from the NH to the PH algorithm, for both exact diagonalization and the SLEPc based iterative solver. For the SLEPc case a five- to six-fold improvement in speed is obtained. As shown in Table 3, the improvements are due to the better scaling exponential (i.e., slope). In the NH implementation, the scaling behavior was similar to that of full diagonalization algorithms, and the performance advantage over diagonalization was mainly due to a smaller prefactor (i.e., a lower intercept in the scaling fit). For the diagonalization case the speed-up from the NH to the PH case is also roughly a factor five, while the improvement is due to both a slightly better scaling and prefactor [57].

Table 3: Slope (A) and intercept (B) parameters from linear regressions of the time-complexity data for the resonant (H) and coupling cases (non-Hermitian (NH) and pseudo-Hermitian (PH)). The upper section reports results obtained on the ISM cluster using the GFortran compiler; these are the data shown in Fig. 1. The lower section presents reproduced benchmarks performed on the University of Valencia cluster (UVC) using Intel compilers and MKL libraries.

| Method | Kind | Slope(A) | Intercept(B) | Kind | Slope(A) | Intercept(B) |
|--------|------|----------|--------------|------|----------|--------------|
| **GFortran Compiler (ISM-CNR Cluster)** | | | | | | |
| LAPACK | H | 2.61 | -7.91 | NH | 3.05 | -7.71 |
| | | | | PH | 2.89 | -8.00 |
| ELPA | H | 2.71 | -8.17 | PH | 2.79 | -7.99 |
| SLEPc Expl. | H | 2.12 | -6.33 | NH | 2.45 | -6.30 |
| | | | | PH | 2.35 | -6.64 |
| Haydock | H | 0.95 | -2.77 | PH | 1.66 | -5.35 |
| **Intel Compiler (University of Valencia Cluster)** | | | | | | |
| LAPACK | H | 2.56 | -7.81 | NH | 3.04 | -8.12 |
| | | | | PH | 2.93 | -8.29 |
| SLEPc Expl. | H | 2.19 | -6.63 | NH | 2.85 | -7.63 |
| | | | | PH | 2.36 | -6.74 |
| Haydock | H | 1.37 | -4.32 | PH | 1.88 | -5.80 |

In Table 3 we also put the extracted slope and intercept from simulations done on a different machine, using Intel compiler with MKL libraries, in order to verify how much these numbers are machine dependent. For a given algorithm, the slope A is expected to be independent of the computational configuration (such as system architecture, compiler, and libraries), as it reflects the intrinsic scaling behavior of the algorithm. In contrast, the intercept B can be seen as the product of two factors: $B_1$, which depends on the configuration, and $B_2$, which is algorithm-dependent. The results confirm that the extracted slope is rather similar between the two machines. In this case the extracted value is $B_1^{ISM}/B_1^{UVC} \approx 0.93$, which suggests that the two clusters have rather similar performances.

Panels (d), (e), and (f) in Fig. 1 and Table 4 illustrate the memory complexity of the studied algorithms, that is, how memory usage increases with matrix size. The following observations can be made: (i) iterative algorithms consume less memory than diagonalization-based algorithms; (ii) memory complexity is similar between LAPACK and ELPA, and between SLEPc and Haydock; and (iii) the PH implementations result in improved memory efficiency vs the NH implementation for the SLEPc algorithms, while causing a slight increase in memory usage for the LAPACK-based approach [58]. Panels (d) and (f) show that the ELPA and LAPACK memory fits overlap almost perfectly. In contrast, a small difference is observed between Haydock and SLEPc, with the Haydock algorithm being slightly more memory-efficient. Overall the memory increase as a function of the matrix size behaves as expected.

Fig. 2 shows the time complexity plots rescaled to reflect the actual size of the coupling matrix. As already observed, iterative solvers are orders faster than diagonalization-based approaches. When the full matrix size is considered, instead of only the resonant block size, it can be seen that the two algorithms (H vs PH) behave rather similar performances. The ELPA implementation is the only one where the PH is clearly faster than the H for all matrix sizes considered here. For the iterative solvers, the Hermitian version exhibits a more favorable slope compared to the pseudo-Hermitian one, and this difference becomes more significant as the matrix size increases.

Table 4: Slope and intercept values for the memory complexity plots shown in Fig. 1(d)–(f), for the resonant (res.) and coupling (cpl.) cases. H – Hermitian, NH – non-Hermitian, PH – pseudo-Hermitian. Expl. - explicit matrix type.

| Method | kind | Slope(A) | Intercept(B) | kind | Slope(A) | Intercept(B) |
|---|---|---|---|---|---|---|
| LAPACK | H | 1.31 | -4.64 | NH | 1.52 | -5.16 |
|  |  |  |  | PH | 1.68 | -5.56 |
| SLEPc Expl. | H | 0.72 | -2.76 | NH | 1.23 | -4.37 |
|  |  |  |  | PH | 1.03 | -3.37 |
| ELPA | H | 1.35 | -4.77 | PH | 1.68 | -5.55 |
| Haydock | H | 0.67 | -2.59 | PH | 0.81 | -3.02 |

370   In summary, this subsection demonstrates that, both for SLEPc and diagonalization, the
371 new PH algorithm greatly improves the time to solution at fixed matrix size compared to the
372 NH algorithm, making the solution of BSE with coupling nearly as efficient as the resonant
373 approximation. Memory usage is also improved for SLEPc, while it remains almost unchanged
374 in the diagonalization case.

### 3.3 CPU scaling. ISM-CNR Cluster

376 Now we analyze MPI scaling of the solvers on CPU. We consider strong scaling with matrices
377 of size $N = 10^4$ and $N = 3 \cdot 10^4$. For the coupling case, this corresponds to matrices of size
378 $2N$. In Fig. 3 time and memory scaling plots are presented, together with efficiency plots. The
379 exponential fit of the data is reported in Table 5. Finally, the extracted parameters from the
380 memory fit are reported in Table 6. In this and next subsection, the term *scaling* is referring to
381 scaling with respect to number of CPUs or GPUs. Moreover, only the pseudo-Hermitian version
382 of the algorithm is considered for the coupling matrices.
383   Fig. 3 shows the execution time of iterative and diagonalizational algorithms (see panels
384 (a) and (b)). However, the efficiency of the SLEPc solver is in general worse than the diago-
385 nalization case. As shown in Fig. 3, panels (c) and (d), efficiency drops from 1 to about 0.4
386 when using up to 16 cores. Beyond that point, efficiency plateaus for the explicit case, while
387 continuing to decline for the shell case. The efficiency of the diagonalization solvers instead
388 ranges in between 0.5 and 0.9. The ELPA library consistently shows the highest efficiency and
389 scaling behaviors (see panels (c) and (d)).

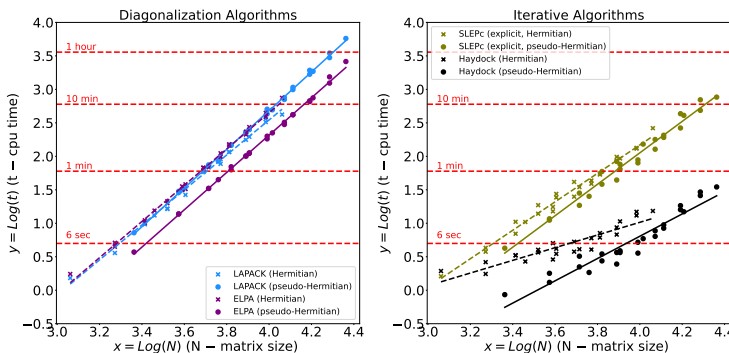

Figure 2: Comparison of time complexity for the studied algorithms, with the x-axis representing the actual matrix size: N for the resonant case and 2N for the coupling case.

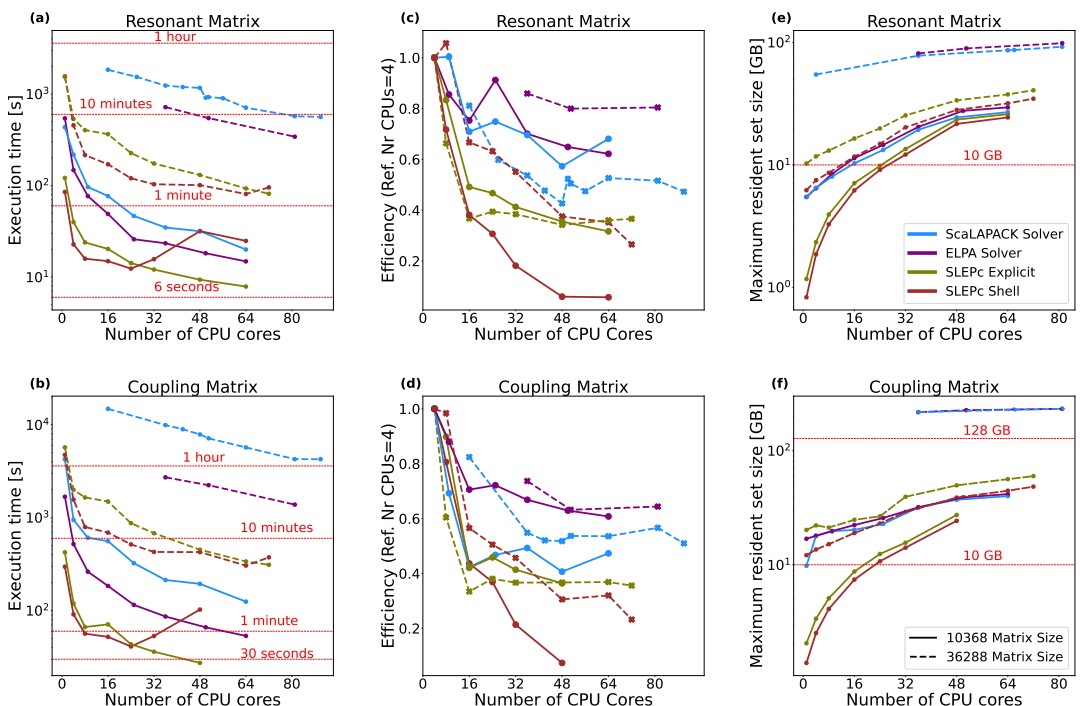

Figure 3: Execution time (panels (a) and (b)), efficiency (panels (c) and (d)), and memory usage (panels (e) and (f)) for different solvers applied to the resonant and coupling cases (pseudo-Hermitian solver) of the BSE eigenvalue problem. CPU case.

Table 5: Exponential fit, $t_i = t_1 \, n_i^p$, of time vs number of CPUs for resonant (res.) and coupling (cpl.) matrices. Where $p = -1$ is the ideal parallelization case. (e) – explicit matrix type, (s) – shell matrix type.

| Solver | $p$ (Fitted Exp.) | | Solver | $p$ (Fitted Exp.) | |
| --- | --- | --- | --- | --- | --- |
| | N=10368 | N=36288 | | N=10368 | N=36288 |
| ScaLAPACK res. | -0.74 | -0.74 | SLEPc res. (e) | -0.63 | -0.65 |
| ScaLAPACK cpl. | -0.79 | -0.77 | SLEPc cpl. (e) | -0.67 | -0.66 |
| ELPA res. | -0.87 | -0.82 | SLEPc res. (s) | -0.25 | -0.66 |
| ELPA cpl. | -0.82 | -0.94 | SLEPc cpl. (s) | -0.36 | -0.60 |

Analyzing in detail different flavours of the iterative and diagonalization solvers we observe the following. For the SLEPc case, the shell matrix algorithm performs very poorly, with a total time which starts to grow when a critical number of MPI tasks is reached. This behaviour is somewhat expected as the shell approach uses a matrix distribution which is optimized for building the kernel. Instead, the explicit approach is more efficient, although not as efficient as full diagonalization. For the diagonalization case, the ELPA library outperforms ScaLAPACK, likely due to its more efficient parallelization. This performance difference can be seen in Fig. 3, panel (a). As the matrix size increases, ELPA's efficiency will plateau in the range of 0.6 − 0.8, depending on the matrix type and size, with larger matrices benefiting more from ELPA's optimization, whereas for ScaLAPACK, the plateau the plateau consistently occurs around 0.4 − 0.5. In the coupling case, panel (b), ELPA shows an even greater performance advantage. This is most likely due to its more optimized implementation of the pseudo-Hermitian solver, compared to ScaLAPACK. Interestingly, in some cases, ELPA's efficiency in the pseudo-Hermitian case is higher than in the Hermitian case. These differences are also reflected in the p values reported in Table 5, where ELPA values are closer to the ideal p = −1, typically around −0.8

Table 6: Distributed (Distr.), duplicated (Dupl.), and parallel overhead (Par.) memory for the CPU case. Memory tracked with `time` command (PETSc tracker in parenthesis). (e) - explicit matrix type, (s) - shell matrix type. All units in [GB].

| Solver | N=10368 | | | N=36288 | | |
|---|---|---|---|---|---|---|
| | Distr. | Dupl. | Par. | Distr. | Dupl. | Par. |
| ScaLAPACK res. | 4.96 | 0.37 | -0.19 | 55.82 | 0.47 | - |
| ScaLAPACK cpl. | 13.54 | 0.45 | 5.96 | 205.98 | 0.34 | - |
| ELPA res. | 4.97 | 0.41 | -0.14 | 68.83 | 0.37 | - |
| ELPA cpl. | 16.15 | 0.42 | -0.47 | 208.09 | 0.32 | - |
| SLEPc res. (e) | 3.68 (0.58) | 0.42 (0.41) | 4.15 | - (9.95) | - (0.44) | - |
| SLEPc cpl. (e) | 4.68 (0.80) | 0.53 (0.51) | 4.82 | - (17.56) | - (0.60) | - |
| SLEPc res. (s) | 0.15 (0.15) | 0.39 (0.40) | -0.39 | - (5.73) | - (0.42) | - |
| SLEPc cpl. (s) | 0.21 (0.22) | 0.46 (0.47) | -1.00 | - (11.31) | - (0.53) | - |

to $-0.9$, followed by ScaLAPACK with values in between $-0.7$ and $-0.8$, and finally the SLEPc explicit solver reaches values in between $-0.6$ and $-0.7$. The SLEPc shell solver, again, shows the worst performance in this comparison.

Duplicated memory remains consistently around 0.4–0.5 GB per CPU across all cases, likely due to internal Yambo operations and is independent of the solver, panels (e) and (f). Its impact decreases with matrix size. In contrast, distributed memory depends on both solver and matrix size. Direct diagonalization shows high memory usage approximately 5–6 times larger than the calculated value for resonant case (see Table 2); and coupling case uses about 3–4 times more memory than the resonant case. SLEPc requires much less memory, enabling single MPI runs, unlike full diagonalization. In the explicit case, memory usage increases notably when transitioning from serial to parallel runs which we attribute to parallel overhead memory (see Table 6). This effect is likely due to additional variable allocations. Reducing this overhead could enhance SLEPc's efficiency.

## 3.4 GPU Scaling. Leonardo Booster

We now analyze performance and MPI scaling of the solvers on GPU. Here we consider matrices of size $N = 10^4$, $N = 3 \cdot 10^4$, and $N = 10^5$, which we refer to as small, medium, and large matrices, respectively. To this end, we focus on the ELPA and SLEPc explicit solvers, since the ScaLAPACK library is not GPU ported, and the SLEPc shell solver is not efficiently scaling. Similar to the CPU case, SLEPc executes an order or two of magnitude faster than ELPA, Fig. 4, panels (a) and (b).

Matrices of size $N \approx 10^4$ are too small for the available resources on GPU, and performances of the solvers are dictated by extra operations which prevents them from scaling efficiently. For the resonant case, both ELPA and SLEPc complete a calculation in under one

Table 7: Exponential fit of time vs number of GPUs, $t_i = t_1 \, n_i^p$. The case $p = -1$ represents the ideal parallelization. The results marked with $^*$ may be influenced by the limited number of data points in the low GPU count range.

| Solver | $p$ (Fitted Exponential) | | |
|---|---|---|---|
| | N=10368 | N=36288 | N=103680 |
| ELPA res. | -0.17 | -0.62 | -0.67 |
| ELPA cpl. | -0.70 | -0.93 | -0.91 |
| SLEPc res. | -0.34 | -0.65 | -0.87 |
| SLEPc cpl. | -0.37 | -0.63 | -0.40$^*$ |

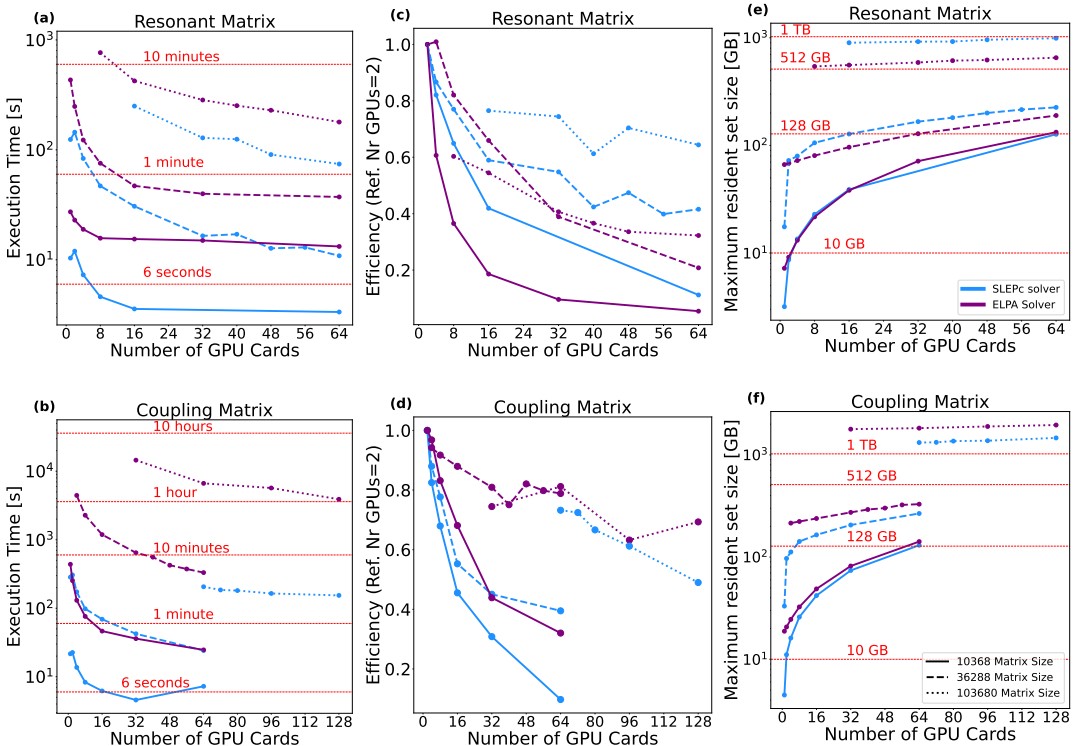

Figure 4: Execution time (panels (a) and (b)), efficiency (panels (c) and (d)), and memory usage (panels (e) and (f)) for different solvers applied to the resonant and coupling cases (pseudo-Hermitian solver) of the BSE eigenvalue problem. GPU case. Only host memory is reported here.

minute on a single GPU (compared to 1–10 minutes on a single CPU). Consequently, performance quickly reaches a plateau: adding more GPUs (beyond 16) yields little to no improvement in execution time. A similar trend is observed for the coupling case, where the plateau begins at 32 GPUs (see Fig. 4). Table 7 shows that for a matrix of size $N \approx 10^4$, the parallelization coefficient p is significantly lower than that for the CPU case. Medium-sized matrices are handled more efficiently. Their parallelization coefficient is higher, and is closer to the values observed in the CPU case (see Tables 5 and 7). For these matrices, a performance plateau is reached around 32 GPUs for the resonant case and around 64 GPUs for the coupling case. Up to the plateau, the scaling is very good: doubling the number of GPUs roughly halves the execution time, though the plateau is reached relatively quickly. For the big matrices the scaling is even better. This is evident from the efficiency plots in Fig. 4, panels (c) and (d), and from the parallelization factors in Table 7. We do not observe any plateau up to the maximum number of GPU used, i.e., 64 (128) for the resonant (coupling) case. Overall we can conclude that, the larger the matrices, the better the scaling and the efficiency.

For the ELPA solver, similarly to the CPU-scaling case, by comparing the resonant and coupling cases, Table 7, we observe that it performs significantly better in the coupling case.

For SLEPc, no significant difference was observed between the resonant and coupling cases. Moreover, oscillations in the execution time can be observed suggesting that numbers of GPU equal to powers of 2 perform better.

Memory usage analysis, panels (e) and (f), and Table 8 shows that duplicated memory on the host remains consistently around 2 GB per used GPU, which is 4 times higher than for the CPU case. The distributed memory is very close to the one obtained in the CPU case, see Table 6. On Leonardo Booster, per each GPU there are 64 GB of memory, and for the cases

Table 8: Distributed (Distr.), duplicated (Dupl.), and parallel overhead (Par.) memory for the GPU case. All units in [GB]. For the largest matrix size, the parallel overhead could not be estimated because single GPU calculations were not feasible due to memory limitations.

| Solver | N=10368 | | | N=36288 | | | N=103680 | |
|---|---|---|---|---|---|---|---|---|
| | Distr. | Dupl. | Par. | Distr. | Dupl. | Par. | Distr. | Dupl. |
| ELPA res. | 5.02 | 2.08 | -0.16 | 64 | 2.00 | -0.17 | 526 | 2.00 |
| ELPA cpl. | 16.51 | 2.03 | -0.38 | 208 | 1.98 | - | 1720 | 1.85 |
| SLEPc res. | 4.13 | 2.11 | 4.36 | 72 | 2.45 | 63.92 | 861 | 1.95 |
| SLEPc cpl. | 6.32 | 2.15 | 5.68 | 102 | 3.45 | 71.89 | 1172 | 2.26 |

of medium and big matrices, this imposes a minimal amount of GPUs needed to perform the simulations. In comparison with the CPU case, on GPUs the SLEPc solver shows similar or even higher memory consumption than the ELPA solver. This is due to the large parallel overhead memory that we are presently investigating. For the GPU implementation, the parallel overhead memory is of a similar order of magnitude to that observed in the CPU case. Specifically, for SLEPc, the parallel overhead memory is large and positive, indicating the allocation of additional variables for parallel runs. Conversely, for ELPA, the parallel overhead is negative, suggesting more efficient parallelization schemes.

# 4   Conclusions

We have shown that solvers based on ELPA, ScaLAPACK and SLEPc libraries can efficiently handle very large BSE matrices, both in the resonant and in the coupling case. This is possible since the interface with the libraries, as implemented in the Yambo code, distributes the memory load over the MPI tasks. Large matrices can then be handled, provided that a sufficient number of MPI tasks is used. Moreover, we have shown a good scaling both on CPUs and GPUs (for ELPA and SLEPc). With the new implementations taking advantage of the pseudo-Hermitian structure of the BSE matrix, it is now possible to handle the coupling matrices almost as efficiently as the resonant matrices, both with iterative solvers based on SLEPc and with diagonalization solvers based on ScaLAPACK and ELPA. The pseudo-Hermitian algorithms used in the coupling case improve both time complexity scaling (reaching values similar to the Hermitian case), and CPU/GPU scaling. For exact diagonalization solvers, the ELPA library outperforms the ScaLAPACK library on CPUs. For the iterative solver based on SLEPc, it is more efficient to redistribute the matrix in memory from the Yambo parallelization scheme to the SLEPc parallelization scheme (explicit solver) at the price of some memory duplication. Re-distribution in memory is also needed for ELPA and ScaLAPACK.

In conclusion, solvers based on external libraries can be reliably used for the Bethe-Salpeter equation (BSE). The situation is somewhat different from other cases, such as the DFT eigenvalue problem, where the solver is one of the most demanding steps. Accordingly, for the DFT case, it might be better to have DFT-specific and more efficient iterative solvers directly inside the code. For the BSE case, the building of the matrix remains, in many cases, the most demanding step on which code developers can focus. Therefore, opting for library-based solvers is a valuable solution, because it allows us to overcome the "solver barrier", which appears for very large matrices, especially when extraction of the eigenvectors is needed. Beyond the libraries considered in this work, we are also interfacing the Yambo code with the Magma library [59], and with the cuSolver and cuSolverMP/cuSolverMG libraries [60]. The biggest advantage for BSE code developers in using libraries is that they outsource the implementa-

tion, MPI distribution, and GPU (or other architecture) porting. This approach is well justified by the performance results presented in this work.

# Acknowledgements

Innovation Study ISOLV-BSE has received funding through the Inno4scale project, which is funded by the European High-Performance Computing Joint Undertaking (JU) under Grant Agreement No 101118139. The JU receives support from the European Union's Horizon Europe Programme. DS acknowledges funding from the MaX "MAterials design at the eXascale" project, co-funded by the European High Performance Computing joint Undertaking (JU) and participating countries (Grant Agreement No. 101093374). DS also acknowledges funding from the European Union's Horizon Europe research and innovation programme under the Marie Sklodowska-Curie grant agreement 101118915 (project TIMES). DS and PM acknowledge the CINECA award under the ISCRA initiative, for the availability of high performance computing resources and support, through the project IsCc6_ISOL-RUN. BM, FA, ER, and JER. were partly supported by grant PID2022-139568NB-I00 funded by MCIN/AEI/10.13039/501100011033 and by "ERDF A way of making Europe". BM was also supported by Universitat Politècnica de València in its PAID-01-23 programme. NM and LW acknowledge the use of HPC facilities of the University of Luxembourg [61].

**Disclosure of Interests.** The authors have no competing interests to declare that are relevant to the content of this article.

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
