# Peer review of "Performance in solving the Hermitian and pseudo-Hermitian Bethe-Salpeter equation with the Yambo code"

_SciPost Physics Codebases_

## Round 1 · Referee Report · Anonymous (Referee 2) · 2025-9-29

Strengths

1- Comparison of different solvers interfaced with the Yambo electronic structure code in terms of runtime, scalability and memory utilization 2- Detailed analysis of results 3- Good readability and writing style

Weaknesses

1- Suitability for journal questionable 2- Structure and conciseness could be improved 3- Details on used solvers are unclear

Report

The manuscript "Performance in solving the Hermitian and pseudo-Hermitian Bethe-Salpeter equation with the Yambo code" presents runtime experiments and results to compare different solvers that can be used to solve the "Bethe-Salpeter eigenvalue problem" arising in condensed matter physics. The electronic structure code "Yambo" is interfaced with solver libraries (Sca)LAPACK and ELPA for direct diagonalization and SLEPc for an iterative solver. The Yambo default solver "Haydock", directly computing the optical spectrum, is also considered. A lot of valuable data is gathered and put into context.

The manuscript does not present a "Codebase" per sé and therefore does not seem to align with the publication criteria of the journal "SciPost Physics Codebases". "Yambo" would be an example of a codebase, as I understand it, but is not presented in its entirety here. Instead, only some new features, i.e. the ability to interface other solver libraries, are evalualed. These additions are not (yet?) implemented in Yambo's production code, but on the development fork "lumen", further limiting their relevance given the journal's scope.

While the manuscript is pleasantly readable, it would benefit from a more clear and concise structure. If one wants to understand, say, one figure, the relevant information (algorithm, library, hardware, ..) seems scattered all over the manuscript.

In particular, perhaps due to the confusing structure of the manuscript, it is unclear which algorithms exactly were used when talking about "non-Hermitian Algorithms" and "pseudo-Hermitian Algorithms". For "non-Hermitian Algorithms", I am assuming it is a general eigensolver that completely ignores the matrices structure, i.e. LAPACK's "zgeev". But this should be stated explicitly. Even more unclear are the ""pseudo-Hermitian Algorithms". Section 2.1.1. implies that the (complex) eigenvalue problem is recast into a real skew-symmetric eigenvalue problem, which is then solved by a special skew-symmetric solver (see references 9 and 32). However, as far as I'm aware, this type of solver is not available in standard (Sca)LAPACK, but only in ELPA. So what exactly was used in the (Sca)LAPACK case? Or was the eigenvalue problem interpreted as a complex generalized Hermitian problem? Then section 2.1.1. is misleading. This should be clarified.

The manuscript should be sent to a journal whose scope better fits its contents, and should improve its clarity.

Requested changes

Some further open questions and points to improve: 1- Footnotes should not be used as sources, but should be actual footnotes or incorporated into the text where it makes sense. 2- The number of computed eigenvalues (100) in the iterative scheme seems arbitrary and specific, but the conclusions drawn are very general. (Example: “As already observed, iterative solvers are orders faster than diagonalization-based approaches.”) They would be more convincing, if the number of computed eigenvalues were varied as well, if the conclusions are supported by algorithmic arguments (instead of observed empirically) or if there was a good reason to choose this number specifically. 3- How was GPU memory measured? The linux time command does not provide this funcitonality. 4- In the hardware overview, details on the node-interconnect are missing, which are relevant in the experiments of Figure 3, as the experiments surpass the 1-node regime when using more than 32 cores. This impact should be addressed.

Recommendation

Accept in alternative Journal (see Report)

---

## Round 1 · Referee Report · Anonymous (Referee 1) · 2025-9-29

Disclosure of Generative AI use

The referee discloses that the following generative AI tools have been used in the preparation of this report:

OpenAI ChatGPT (GPT-5), used solely to check the English grammar of the report

Report

The present manuscript is clearly written, technically sound, and addresses a computational bottleneck in Bethe-Salpeter equation. This is relevant for condensed matter physics and materials science. The benchmarking against CPU/GPU clusters, as well as the comparison between Hermitian, non-Hermitian, and pseudo-Hermitian formulations, is accurate. The work will be valuable for the community, especially Yambo users. Overall the manuscript is technically strong, with solid contributions.

I support the publication of this work, and would like the authors to reply to the following comments:

1) While the manuscript focuses on performance, it would benefit from a short comment on some scientific applications. For example: what system sizes or particular physical phenomena (e.g. excitons in specific 2D materials, Rydberg states, etc.) become accessible with these advances? Some readers may miss why a BSE matrix with size 10^5 matters physically, or what kind of excitonic spectra these calculations are enabling.

2) As far as I can see, the benchmarks are primarily performed using the BSE Hamiltonian of CrI3. Could the authors clarify to what extent the conclusions obtained from this material are expected to be general? In particular, are there material dependent features (e.g. band dispersion, screening etc.) that might affect the efficiency or scaling?

Recommendation

Publish (easily meets expectations and criteria for this Journal; among top 50%)

---

## Round 2 · Author Response

In short, our paper presents both a novel algorithm for solving the Bethe-Salpeter equation and a more efficient reimplementation of already existing algorithms. This is fully in line with the scope of the journal, as described in the About section of Physics Codebases (https://scipost.org/SciPostPhysCodeb/about). A more detailed explanation is available in the corresponding section of the list of changes and remarks for editors.

---

## Round 2 · List of Changes

The structure is the following: First a general overview of each of referees reply. And after, each of the points raised by referees was copied into this text as a question, to which we reply here, and to which we make changes in the article as well.
In the end of this reply, Appendix B, there will be the list with the newly introduced line numbers and the remarks they address.
—----------------
Report #1 by Anonymous (Referee 1) on 2025-9-29 (Invited Report)
—----------------
"The present manuscript is clearly written, technically sound, and addresses a computational bottleneck in Bethe-Salpeter equation. This is relevant for condensed matter physics and materials science. The benchmarking against CPU/GPU clusters, as well as the comparison between Hermitian, non-Hermitian, and pseudo-Hermitian formulations, is accurate. The work will be valuable for the community, especially Yambo users. Overall the manuscript is technically strong, with solid contributions.
I support the publication of this work, and would like the authors to reply to the following comments: "
—----------------
We thank the referee for his positive feedback and we are glad to address his remarks.
—----------------
Remark 1.
“While the manuscript focuses on performance, it would benefit from a short comment on some scientific applications. For example: what system sizes or particular physical phenomena (e.g. excitons in specific 2D materials, Rydberg states, etc.) become accessible with these advances?”
Answer
We specified this better in the introduction of the manuscript, by adding the following sentence: “It explicitly accounts for electron–hole interactions, allowing for an accurate description of excitonic effects, and thereby enabling reliable prediction of optical properties in a wide range of materials, including 2D materials [1,2], wide band gap insulators and transition metal oxides [3,4], semi-conductors and magnetic materials [5,6]. It was also recently used to compute magnons in magnetic materials [7].”
—----------------
Remark 2.
"Some readers may miss why a BSE matrix with size 10^5 matters physically, or what kind of excitonic spectra these calculations are enabling."
Answer
Increasing the size of the BSE matrix is essential to achieve convergence of excitonic energies and to accurately describe exciton (and magnons) physics in advanced materials. More specifically, matrices with size of the order 10^4 - 10^5 can be easily reached when studying 2D heterostructures [8,9], systems with defects [10,11], systems that involve molecular interactions such as functionalized materials (e.g., functionalized graphene) or systems containing impurities. When the coupling term is included, the matrix size for the same system, using the same number of bands, will increase four-fold. In certain cases, neglecting the coupling term can lead to qualitatively incorrect results. For instance, the coupling term is known to be essential when studying localized states which are important in some of the applications mentioned above. As a result, all of them require the full BSE treatment. We have added a paragraph in the text to clarify for readers the importance of matrix size in obtaining physically reliable results.
—----------------
Remark 3.
"As far as I can see, the benchmarks are primarily performed using the BSE Hamiltonian of CrI3. Could the authors clarify to what extent the conclusions obtained from this material are expected to be general? In particular, are there material dependent features (e.g. band dispersion, screening etc.) that might affect the efficiency or scaling?"
Answer
The specific material used and its properties (e.g. band dispersion, screening etc.) affects the timing needed to construct the BSE kernel. As mentioned in the text, we do not use the actual BSE kernel but a TDDFT kernel, which is computationally less demanding to generate. Although the TDDFT kernel represents a different physical approximation, the numerical procedure used to solve it is identical to that of the BSE kernel, which is the central focus of this paper. It’s important to mention that, in any case, the resulting matrix is dense.
Once the matrix is constructed, the problem reduces to an eigenvalue problem, and therefore the computational complexity is determined solely by the matrix size. The diagonalization algorithm scales as N3 and the time to solution does not depend on the matrix properties. For the iterative SLEPc solver, the time required for a single iteration is also independent of the matrix properties, while the number of iterations needed might be affected by the matrix properties. For instance, if the matrix has clustered eigenvalues, this will hinder convergence, resulting in a larger number of iterations. However, this does not usually occur in BSE applications, and the chosen matrix gives a good representation of the needed time to solution for a wide range of cases.
The quantities of interest for our study are the computational time required to obtain a solution, efficiency and parallelization of the employed algorithms. We have further clarified this point in the manuscript to emphasize that the choice of material does not affect the conclusions presented here.
—----------------
Report #2 by Anonymous (Referee 2) on 2025-9-29 (Invited Report)
—----------------
"The manuscript "Performance in solving the Hermitian and pseudo-Hermitian Bethe-Salpeter equation with the Yambo code" presents runtime experiments and results to compare different solvers that can be used to solve the "Bethe-Salpeter eigenvalue problem" arising in condensed matter physics. The electronic structure code "Yambo" is interfaced with solver libraries (Sca)LAPACK and ELPA for direct diagonalization and SLEPc for an iterative solver. The Yambo default solver "Haydock", directly computing the optical spectrum, is also considered. A lot of valuable data is gathered and put into context."
—----------------
We thank the referee.
—----------------
"The manuscript does not present a "Codebase" per sé and therefore does not seem to align with the publication criteria of the journal "SciPost Physics Codebases". "Yambo" would be an example of a codebase, as I understand it, but is not presented in its entirety here. Instead, only some new features, i.e. the ability to interface other solver libraries, are evaluated. These additions are not (yet?) implemented in Yambo's production code, but on the development fork "lumen", further limiting their relevance given the journal's scope."
—----------------
We replied in detail for the editor in charge on why our paper meets the criteria to be published in "SciPost Physics Codebases". We also changed the title of the manuscript to better stress this point. A copy of the reply to the editor is presented in the Appendix A.
—----------------
Remark 4.
"While the manuscript is pleasantly readable, it would benefit from a more clear and concise structure. If one wants to understand, say, one figure, the relevant information (algorithm, library, hardware, ..) seems scattered all over the manuscript.
In particular, perhaps due to the confusing structure of the manuscript, it is unclear which algorithms exactly were used when talking about "non-Hermitian Algorithms" and "pseudo-Hermitian Algorithms""
Answer
We are not sure what the referee means here. The information about the libraries, hardware, compiler, software, is presented in Table 1. At the beginning of Chapter 3, we define the terms non-Hermitian algorithm and pseudo-Hermitian algorithm. A non-Hermitian algorithm refers to ELPA, LAPACK, or SLEPc solvers that do not exploit the internal structure of the matrix. Details regarding their specific implementations can be found in the corresponding references cited in the Introduction. In contrast, a pseudo-Hermitian algorithm explicitly accounts for the pseudo-Hermitian structure of the matrix. This structure arises when the coupling term is included, and we refer to the whole matrix as “coupling matrix”. The diagonalization in this case follows the scheme described in Section 2.
—----------------
Remark 5.
"Section 2.1.1. implies that the (complex) eigenvalue problem is recast into a real skew-symmetric eigenvalue problem, which is then solved by a special skew-symmetric solver (see references 9 and 32). However, as far as I'm aware, this type of solver is not available in standard (Sca)LAPACK, but only in ELPA. So what exactly was used in the (Sca)LAPACK case? "
Answer
The referee is right, (Sca)LAPACK does not provide a skew-symmetric solver. This was not clearly explained and we clarified this in the manuscript. We have added the following paragraph to the text. “Since ScaLAPACK does not provide a dedicated solver for real skew-symmetric matrices, the tridiagonalization routines designed for symmetric matrices can be slightly modified to accommodate them, as both share similar computational structures. For the present implementation, we employ ScaLAPACK's Hermitian solver by passing the matrix $-iW$, which converts the real skew-symmetric matrix $W$ into a Hermitian form. In future releases, we plan to modify ScaLAPACK's real tridiagonalization routines to directly handle the skew-symmetric matrix $W$, thereby eliminating the need for this transformation”.
—----------------
Remark 6.
"Footnotes should not be used as sources, but should be actual footnotes or incorporated into the text where it makes sense. "
Answer
We have encountered both practices across different journals, using footnotes either within the main text or as part of the reference list. In this manuscript, we have followed the latter approach. However, we are happy to adjust the format according to the journal’s preferred style and leave this decision to the editor’s discretion.
—----------------
Remark 7.
"The number of computed eigenvalues (100) in the iterative scheme seems arbitrary and specific, but the conclusions drawn are very general."
Answer
In iterative schemes, the number of computed eigenvalues is typically selected to capture the first excitonic peak. In most cases, this number ranges from 20 to 30. Therefore, in our study, we chose to compute 100 eigenvalues, which is more than sufficient to describe the relevant low-energy excitations. When a larger number of excitations or eigenvalues is required, a full diagonalization solver becomes the more appropriate choice. For the purposes of this work, we do not find it necessary to perform a convergence study on the number of eigenvalues obtained with the iterative solver. A clarifying comment has been added to the manuscript to explain the rationale behind choosing 100 eigenvalues.
—----------------
Remark 8.
"How was GPU memory measured? The linux time command does not provide this functionality."
Answer
We did not measure the GPU memory usage. We measured only the host memory, while running on the simulations on a GPU card. Due to the way the algorithms are GPU ported, the host memory is an upper bound to the actual memory used in the card. Also we specified in the text that the time command measures the memory on the host.
—----------------
Remark 9.
"In the hardware overview, details on the node-interconnect are missing, which are relevant in the experiments of Figure 3, as the experiments surpass the 1-node regime when using more than 32 cores. This impact should be addressed."
Answer
We have added in Table 1 the information about the node interconnect.
—----------------
References
[1] S. Haastrup et al., The Computational 2D Materials Database: high-throughput modeling and discovery of atomically thin crystals, 2d Mater 5, 042002 (2018).
[2] P. Lechifflart, F. Paleari, and C. Attaccalite, Excitons under strain: light absorption and emission in strained hexagonal boron nitride, SciPost Physics 12, 145 (2022).
[3] M. Gatti and F. Sottile, Exciton dispersion from first principles, Phys Rev B 88, 155113 (2013).
[4] L. Varrassi, P. Liu, Z. E. Yavas, M. Bokdam, G. Kresse, and C. Franchini, Optical and excitonic properties of transition metal oxide perovskites by the Bethe-Salpeter equation, Phys Rev Mater 5, 074601 (2021).
[5] S. Albrecht, L. Reining, R. Del Sole, and G. Onida, Ab Initio Calculation of Excitonic Effects in the Optical Spectra of Semiconductors, Phys Rev Lett 80, 4510 (1998).
[6] M. Wu, Z. Li, T. Cao, and S. G. Louie, Physical origin of giant excitonic and magneto-optical responses in two-dimensional ferromagnetic insulators, Nat Commun 10, 2371 (2019).
[7] A. Esquembre-Kučukalić, K. B. Le, A. García-Cristóbal, M. Bernardi, D. Sangalli, and A. Molina-Sánchez, Magnons in chromium trihalides from ab initio Bethe-Salpeter equation, (2025).
[8] C. Bacaksiz, A. Dominguez, A. Rubio, R. T. Senger, and H. Sahin, ℎ-AlN-Mg(OH)2 van der Waals bilayer heterostructure: Tuning the excitonic characteristics, Phys Rev B 95, 075423 (2017).
[9] R. Reho, A. R. Botello-Méndez, D. Sangalli, M. J. Verstraete, and Z. Zanolli, Excitonic response in transition metal dichalcogenide heterostructures from first principles: Impact of stacking, twisting, and interlayer distance, Phys Rev B 110, 035118 (2024).
[10] A. Kirchhoff, T. Deilmann, and M. Rohlfing, Excited-state geometry relaxation of point defects in monolayer hexagonal boron nitride, Phys Rev B 109, 085127 (2024).
[11] R. R. Del Grande and D. A. Strubbe, How to choose efficiently the size of the Bethe-Salpeter equation Hamiltonian for accurate exciton calculations on supercells, Phys Rev B 112, 165118 (2025).
—----------------—----------------
Appendix A
A copy of the reply to the editor, to answer the concern of the referee #2 regarding the choice of the journal.
—----------------—----------------
Dear Editor,
Please find attached a revised version of our manuscript, with the new title:
“Solvers for the Hermitian and the pseudo-Hermitian Bethe-Salpeter equation in the Yambo code: implementation and performances”
To be considered for publication in SciPost Physics Codebases.
The judgment of both referees is to publish the manuscript.
The first referee states “The present manuscript is clearly written, technically sound, and addresses a computational bottleneck in Bethe-Salpeter equation” and recommends “Publish (easily meets expectations and criteria for this Journal; among top 50%)”.
The second referee also judges well the manuscript “the manuscript is pleasantly readable”, and his conclusion is to “Accept” the manuscript. However, the second referee also suggests “in alternative journal”, since “The manuscript does not present a "Codebase" per sé and therefore does not seem to align with the publication criteria of the journal "SciPost Physics Codebases".
Here we agree with the editor that the manuscript is not suitable for other journals in "SciPost Physics”, and we respectfully disagree with the second referee on the conclusion that our manuscript is not suitable for "SciPost Physics Codebases"
We decided to submit to "SciPost Physics Codebases" because, as stated in the About section on the journal’s website, “SciPost Physics Codebases is a new-generation journal for computer codes and algorithms of relevance to research in Physics.” Later, in the same page, the scope of the journal is also better highlighted:
SciPost Physics Codebases publishes outstanding-quality Codebases relevant to all specializations in Computational, Experimental and Theoretical Physics.
Examples of publishable Codebases include:
Novel algorithms
Significant and original reimplementations of well-known algorithms
Ports of existing codebases to new languages and platforms
Libraries providing new or significantly improved components or interfaces which enhance the capability, performance, or productivity of scientific software.
Our article presents the implementation and the performances of 3 different algorithms:
a novel algorithm for solving the Bethe–Salpeter Equation with coupling, using an iterative approach which takes advantage of the pseudo-hermitian structure of the matrix (SLEPc with coupling);
the re-implementation of a recent algorithm proposed in the literature for efficiently solving the Bethe–Salpeter Equation with coupling, via rewriting it in the form of a skew symmetric matrix ;
finally a novel efficient interface with the ELPA and ScaLAPACK libraries for solving the Bethe–Salpeter Equation without coupling, which enhances the capability, performance and productivity of the Yambo code
All the above discussed algorithms are released as part of the “BSE component” of the Yambo code (quoting wikipedia a “codebase” also refers to a “software component”) on the gitlab platform inside the lumen repository. Indeed, the “SLEPc with coupling” algorithm is the output of an “Inno4Scale grant” (Innovative Algorithms for exascale applications). As we have shown, our new implementation provides a significant improvement in computational efficiency of the Yambo code in solving the BSE.
Having an efficient BSE solver is crucial to be able to compute optical properties of advanced materials, and fully justifies having a new release of the code. BSE is actually one of the two main components (together with GW) of the Yambo code. We better clarified the use cases for the Yambo code in the revised version of the manuscript.
For all these reasons we are convinced that is fully within the scope of SciPost Physics Codebases journal. The new title and the changes in the abstract are meant to better highlight these aspects.
A detailed response to all the other comments from the referees is also provided, where we also address the relevance of the “BSE component” of the Yambo code for the community.
—----------------—----------------
Appendix B
Lines changed in the new document.
Lines 22-26 - Remark 1
Lines 96-107 - Remark 2
Lines 175-181 - Remark 5
Lines 263-269 - Remark 7
Line 290 - Remark 8 (Partially)
Lines 302-312 - Remark Remark 3
Table 1 - Remark 9

---

## Editorial Decision

unknown